# Survival and Genome Diversity of *Vibrio parahaemolyticus* Isolated from Edible Aquatic Animals

**Dingxiang Xu [1], Xu Peng [2] , Lu Xie [3] and Lanming Chen [1],***

[1]  Key Laboratory of Quality and Safety Risk Assessment for Aquatic Products on Storage and Preservation (Shanghai), Ministry of Agriculture and Rural Affairs of China, College of Food Science and Technology, Shanghai Ocean University, Shanghai 201306, China; m190300759@st.shou.edu.cn

[2]  Department of Biology, University of Copenhagen, DK 2200 Copenhagen N, Denmark; peng@bio.ku.dk

[3]  Institute for Genome and Bioinformatics, Shanghai Institute for Biomedical and Pharmaceutical Technologies, Shanghai 201203, China; luxiex2017@outlook.com

*  Correspondence: lmchen@shou.edu.cn

**Abstract:** *Vibrio parahaemolyticus* can cause acute gastroenteritis, wound infection, and septicemia in humans. The waterborne bacterium is frequently isolated from aquatic products worldwide. Nevertheless, little information in genome evolution of *V. parahaemolyticus* isolated from aquatic animals is yet available. Here we overcome this limitation by specifying six *V. parahaemolyticus* isolates recovered from edible shellfish, fish, and crustacean. Most isolates with multiple resistance phenotypes grew optimally at 3% NaCl and pH 8.5. Draft genome sequences of the six *V. parahaemolyticus* isolates (4,937,042 bp to 5,067,778 bp) were determined using the Illumina Hiseq $\times$ 10 sequencing platform. Comparative genomic analyses revealed 4622 to 4791 predicted protein-encoding genes, of which 1064 to 1107 were of unknown function. Various mobile genetic elements (MGEs) were identified in the *V. parahaemolyticus* genomes, including genome islands (n = 5 to 9), prophage gene clusters (n = 0 to 2), integrons (n = 1 to 11), and insertion sequences (n = 0 to 3). A number of antibiotic-resistant (n = 17 to 20), virulence-associated (n = 77 to 79), and strain-specific (n = 131 to 287) genes were also identified, indicating possible horizontal gene transfer via the MGEs and considerable genome variation in the *V. parahaemolyticus* isolates. Altogether, the results of this study fill prior gaps in our knowledge of the genome evolution of *V. parahaemolyticus*, as isolated from edible aquatic animals.

**Keywords:** *Vibrio parahaemolyticus*; genome evolution; mobile genetic elements; virulence; antibiotic resistance; aquatic animals

## 1. Introduction

*Vibrio parahaemolyticus* is a Gram-negative bacterium that resides in warm estuarine and marine environments worldwide [1,2]. The bacterium was originally identified in 1950 in Osaka, Japan, where an outbreak of acute gastroenteritis, caused by contaminated semi-dried juvenile sardines, sickened 272 and killed 20 people [3]. It has been estimated that 80,000 cases of *V. parahaemolyticus* infection occur every year; hospitalization and mortality rates thereof were 22% and 1%, respectively [4]. In China, *V. parahaemolyticus* has been identified as the leading cause of foodborne outbreaks, especially in coastal regions [5,6]. Toxic *V. parahaemolyticus* isolates have been found to carry two main virulence genes *tdh* and *trh*, encoding a thermostable direct hemolysin (TDH) and a TDH-related hemolysin, respectively [7,8].

*V. parahaemolyticus* is frequently isolated from aquatic products worldwide [5,9]. In tandem with rapidly expanding aquaculture, the inappropriate use of antimicrobial agents is challenging food safety systems and human health, particularly in developing countries [10]. Antibiotic residues have been detected in aquatic environments and aquatic products [11–13]. For instance, Ni et al. recently reported that 10 commonly used antimicrobial drugs were detected in 14 species of edible aquatic animal samples (n = 108)

that were collected in large aquatic markets in the summers of 2018 and 2019 in Shanghai, China, showing an overall detection frequency of 61.3% [11]. Antimicrobial-resistant *V. parahaemolyticus* strains recovered from aquatic products have also been reported worldwide [12,13]. For example, Su et al. reported a total of 561 *V. parahaemolyticus* isolates recovered from 23 species of edible aquatic animals. High percentages of resistance to ampicillin (AMP) (93.0%), rifampin (RIF) (82.9%), streptomycin (SM) (75.4%), and kanamycin (KAN) (50.1%) were observed among the isolates [13]. Meanwhile, a high incidence of tolerance to heavy metals such as mercury (Hg) (74.7%) and zinc (Zn) (56.2%) was also observed [13]. Heavy metals can co-select for resistance to clinically important antibiotics in ecosystems [2].

Aquatic ecosystems are known to be antibiotic-resistant gene pools [14], where the dissemination of resistance genes could partially be attributed to horizontal gene transfer (HGT), mediated via mobile genetic elements (MGEs) such as integrons, transposons, and phages [15,16]. HGT in prokaryotes is mainly achieved through three mechanisms, including transformation, conjugation, and transduction [17–19]. HGT mediated by MGEs in *V. parahaemolyticus* isolates recovered from seawater in Dalian and Donggang in China was observed, which might enable *V. parahaemolyticus* to adapt to the environment [1].

In concert with increasing breakthroughs in genome sequencing technology [20], the number of genome sequencing projects focused on *V. parahaemolyticus* isolates has clearly increased. A total of 1718 *V. parahaemolyticus* isolates have been sequenced (GenBank database, https://www.ncbi.nlm.nih.gov/, accessed on 17 September 2021), among which, complete genome sequences of 63 *V. parahaemolyticus* isolates are now available. Nevertheless, few studies on comparative genome analyses of these *V. parahaemolyticus* genomes have yet been conducted [3,8,21,22].

In our previous studies, a number of *V. parahaemolyticus* strains were isolated from various commonly consumed aquatic products and characterized [8,13,23]. Of these, the complete genome sequence of *V. parahaemolyticus* CHN25 strain, isolated from shrimp, has been determined using the 454-pyrosequencing technique [24]. MGEs, including insertion sequences (ISs) (n = 5), prophages (n = 5), and integrative and conjugative elements (ICEs) (n = 1), were identified in *V. parahaemolyticus* CHN25 genome [24]. Based on our previous studies, we therefore asked what genome features of the *V. parahaemolyticus* isolates might originate in other species of edible aquatic animals. Thus, the major objectives of this study were (1) to examine the survival of *V. parahaemolyticus* isolates recovered from six species of edible shellfish, fish, and crustaceans at different pH and NaCl conditions; (2) to determine draft genome sequences of the six *V. parahaemolyticus* isolates showing multiple resistance phenotypes using the Illumina Hiseq × 10 (Illumina, San Diego, CA, USA) sequencing technique; (3) to identify MGEs and virulence- and resistance-related genes in the *V. parahaemolyticus* genomes. To the best of our knowledge, this study was the first to perform comparative genomics analysis of *V. parahaemolyticus* isolates recovered from the six species of edible aquatic animals. The results in this study will enrich genome data and fill gaps in our knowledge of genome evolution of *V. parahaemolyticus* isolates originating in edible aquatic animals.

## 2. Materials and Methods

### 2.1. V. parahaemolyticus Isolates and Culture Conditions

*V. parahaemolyticus* N3-33, N4-46, N8-42, L7-7, Q8-15, and N1-22 isolates (Table 1) were recovered from three species of shellfish: *Paphia undulate*, *Perna viridis*, *Mactra veneriformis*; two species of fish: *Aristichthys nobilis*, *Carassius auratu*; and one species of crustacean: *Litopenaeus vannamei*, respectively [13]. The isolates were stored at −80 °C in a freezer in our laboratory at Shanghai Ocean University, Shanghai, China. *V. parahaemolyticus* isolates were individually inoculated in Tryptone Soya Broth (TSB) medium (Beijing Land Bridge Technology, Beijing, China) and incubated with shaking at 175 rpm at 37 °C. The pH and salinity of the TSB medium were adjusted as described previously [23]. Growth curves

were determined using a Multimode Microplate Reader (BioTek Instruments, Winooski, VT, USA), and the $OD_{600nm}$ value was used as a parameter for bacterial biomass [25].

**Table 1.** Genome features of the six *V. parahaemolyticus* isolates originating in edible aquatic animals.

| Genome Feature | *V. parahaemolyticus* Isolate | | | | | |
|---|---|---|---|---|---|---|
| | L7-7 | N1-22 | N3-33 | N4-46 | N8-42 | Q8-15 |
| Genome size (bp) | 4,937,042 | 5,067,778 | 5,018,989 | 5,049,514 | 5,010,476 | 4,993,599 |
| G + C (%) | 45.36 | 45.20 | 45.32 | 45.29 | 45.39 | 45.34 |
| DNA Scaffold | 54 | 58 | 60 | 77 | 61 | 36 |
| Total predicted gene | 4749 | 4919 | 4815 | 4878 | 4832 | 4756 |
| Protein-coding gene | 4624 | 4791 | 4707 | 4768 | 4706 | 4622 |
| RNA gene | 125 | 128 | 108 | 110 | 126 | 134 |
| Genes assigned to COG | 3862 | 3892 | 3852 | 3886 | 3875 | 3854 |
| Genes with unknown function | 1064 | 1094 | 1086 | 1107 | 1102 | 1093 |
| GI | 6 | 9 | 6 | 6 | 7 | 5 |
| Prophage gene cluster | 0 | 1 | 0 | 1 | 2 | 1 |
| CRISPR-Cas | 0 | 2 | 3 | 0 | 0 | 1 |
| Integron | 2 | 1 | 5 | 11 | 8 | 6 |
| IS | 3 | 0 | 1 | 0 | 1 | 1 |
| 16S rDNA sequence accession no. | MW386441 | MW386442 | MW386445 | MW386446 | MW386450 | MW386453 |

### 2.2. Genomic DNA Preparation

Bacterial cells grown to the logarithmic growth phase in TSB medium (3% NaCl, pH 8.5) at 37 °C were harvested by centrifugation at $2700 \times g$ for 10 min at 4 °C. Genomic DNA was prepared using a TIANamp Bacteria DNA Kit (Tiangen Biochemical Technology Co Ltd., Beijing, China) according to the instructions of the manufacturer. Extracted DNA samples were analyzed by agarose gel electrophoresis, and DNA concentration and purity ($A_{260}/A_{280}$) were measured as described previously [26]. Only pure genomic DNA samples (a 260/280 nm absorbance ratio of 1.8–2.0) were used for genome sequencing.

### 2.3. PCR Assay

The primers (Table S1) were synthesized by Sangon Biotech (Shanghai, China) Co., Ltd. (Shanghai, China); 20 µL of PCR reaction system and reaction condition were the same as described previously [13]. PCR reactions were performed using Mastercycler Rpro PCR thermal cycler (Eppendorf, Hamburg, Germany). Amplicons were analyzed by agarose gel electrophoresis, then visualized and recorded as described previously [13]. PCR products were sequenced by Sangon (Shanghai, China).

### 2.4. Genome Sequencing, Assembly, and Annotation

Whole-genome sequencing of the *V. parahaemolyticus* isolates was conducted at Shanghai Majorbio Bio-Pharm Technology Co., Ltd. (Shanghai, China) using the Illumina Hiseq × 10 (Illumina, San Diego, CA, USA) platform. The PE150 (pair-end) sequencing (insert size: 400 bp) was performed with a read length of 150 bp. Low-quality sequence filtering and high-quality sequence assembly were performed using SOAPdenovo (version 2.04) software [27]. The processes included: (1) removing adapter sequences in sequencing reads; (2) shearing and removing non-A, G, C and T bases at the 5′ end of reads; (3) trimming the ends of reads with sequencing quality values less than Q20; (4) removing reads with 10% of N base; (5) discarding small fragments less than 25 bp after the trimming of adapter and low-quality sequences. The coding sequences (CDSs) were predicted using Glimmer (version 3.02) software [28]; rRNA genes were predicted using Barrnap tool (https://github.com/tseemann/barrnap, accessed on 11 April 2021); and tRNA genes were detected using tRNAscan-SE (version 2.0) software [29]. The programs were run with default parameters [27–29].

Functional assignments were inferred using standalone Basic Local Alignment Search Tool (BLAST) (http://www.ncbi.nlm.nih.gov/BLAST, accessed on 11 April 2021) searches against the non-redundant protein database of the National Center for Biotechnology Information (NCBI, http://www.ncbi.nlm.nih.gov, accessed on 11 April 2021) and the clusters of orthologous groups (COG) database [30]. Each predicted protein was annotated if it met the same criteria as described previously [26]. Each gene was also functionally classified by assigning a COG number with 80% identity and 90% coverage at E $\leq 1 \times 10^{-5}$. If the COG did not have a specially appointed function with CDS, the comment was function unknown. The virulence factor database (http://www.mgc.ac.cn/VFs, accessed on 11 April 2021) and antibiotic resistance gene database (http://arpcard.Mcmaster.ca, accessed on 11 April 2021) were used to detect virulence- and antibiotic-resistant-associated genes, respectively.

### 2.5. Prediction of MGEs

The assembled contigs of each genome were subjected to the following MGEs analyses. Genome islands (GIs) were predicted using IslandViewer software (version 1.2) [31]. Prophages were predicted using Phage_Finder software [32]. CRISPR-Cas systems were predicted using Mined software (version 3) and CRISPRtyper (https://cctyper.crispr.dk/#/submit accessed on 11 April 2021) [33]. Integrons were predicted using Integron_Finder software (version 2.0) [34]. Insertion sequences (ISs) were predicted using ISEScan software (version 1.7.2.1) [35]. The programs were run with default parameters [31–35]. ICEs were predicted by searching ICEs and chromosomal junction sequences within the *prfC* gene, and conserved core genes of SXT/R391 ICEs as described previously [36].

### 2.6. Comparative Genome Analysis

The core genes are the set of genes that encode orthologous proteins in all the analyzed genomes, while the pan-genes are the set of all genes that are present in all the analyzed genomes. Orthologous genes were analyzed using OrthoMCL software [37]. Orthologous proteins were assigned only for proteins sharing both 60% amino acid identity and 80% sequence coverage, and those with lower than 30% or no hits were assigned as strain-specific genes at E $\leq 1 \times 10^{-5}$.

To construct a phylogenetic tree, amino acid data sets of single-copy orthologs that were present in all the analyzed genomes were inferred and aligned using OrthoFinder (version 2.2.6) and MUSCLE software [38,39]. The maximum likelihood approach was used to construct a phylogenomic tree using RAxML (version 8) software [40] with 1000 bootstrap replications, and the cutoff threshold for bootstrap values was above 50%.

## 3. Results

### 3.1. Genotypes and Phenotypes of the Six V. parahaemolyticus Isolates

The *V. parahaemolyticus* L7-7, N1-22, N3-33, N4-46, N8-42, and Q8-15 isolates (Table 1), which were recovered from the six species of edible aquatic animals, were confirmed by 16S rDNA gene sequencing and analysis. The obtained sequences were deposited in the Genbank database under the accession numbers shown in Table 1.

All the *V. parahaemolyticus* isolates were detected as negative for the toxins-encoding genes *tdh* and *trh*. However, they were resistant to ampicillin (AMP) and rifampin (RIF), and most to tetracycline (TET) (n = 5). Tolerance to $Zn^{2+}$ and $Hg^{2+}$ was common among the *V. parahaemolyticus* isolates [13].

### 3.2. Survival of the Six V. parahaemolyticus Isolates at Different pH and NaCl Conditions

Human acidic stomach environment (usually 1.0 to 3.0, but it can rise to more than 6.0 after eating) challenges *V. parahaemolyticus* survival and infection in humans [23]. Therefore, we examined the survival of the six *V. parahaemolyticus* isolates at different pH conditions (pH 6.0–8.5) in TSB medium (3% NaCl) at 37 °C. As illustrated in Figure 1A–F), in the acidic conditions (pH 6.0–6.5), the growth of all the *V. parahaemolyticus* isolates was severely

inhibited, and the maximum $OD_{600nm}$ values after reaching the stationary phase (SP) ranged from 0.79 to 0.92; in the neutral condition (pH 7.0), *V. parahaemolyticus* N1-22 isolated from crustacean *L. vannamei* grew vigorously ($OD_{600nm}$ = 1.59) (Figure 1B), whereas the other isolates still grew poorly (Figure 1A,C–F); in alkaline conditions (pH 7.5 to 8.5), all the *V. parahaemolyticus* isolates grew well with the maximum biomass at SP ranging from 1.21 to 1.49 at pH 8.5 (Figure 1).

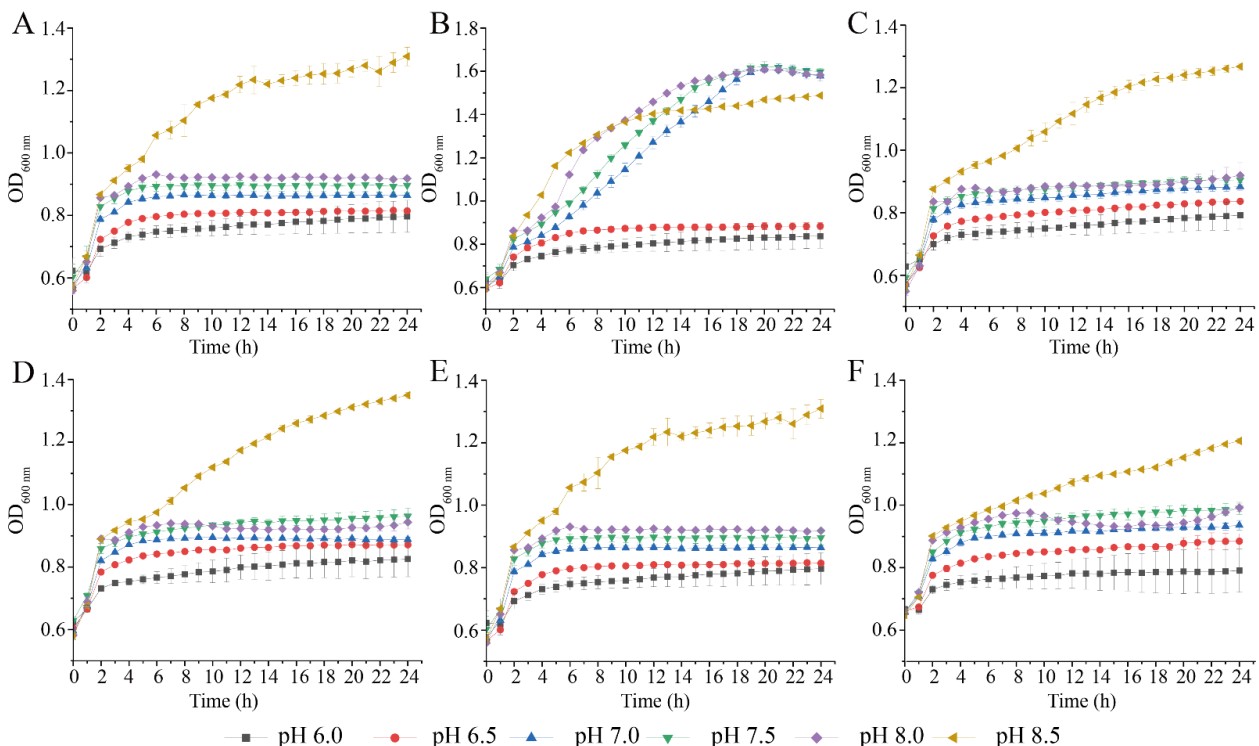

**Figure 1.** Survival of six *V. parahaemolyticus* isolates in different pH conditions. The isolates were incubated in TSB medium (3% NaCl) at 37 °C. (**A–F**): *V. parahaemolyticus* L7-7, N1-22, N3-33, N4-46, N8-42, and Q8-15, respectively.

The survival of *V. parahaemolyticus* in aquatic environments is often challenged by salinity [21]. Therefore, growth curves of the six *V. parahaemolyticus* isolates in TSB (pH 8.5) at 37 °C were determined at different NaCl concentrations (0.5% to 5%). As illustrated in Figure 2A–F, the growth of all the *V. parahaemolyticus* isolates was inhibited at 0.5% NaCl. Although these isolates grew exuberantly at 1% to 5% NaCl, the highest biomass was observed when the isolates grew at 3% NaCl, showing the maximum $OD_{600nm}$ values at SP ranging from 1.35 to 1.44. The results demonstrated that the six *V. parahaemolyticus* isolates of edible aquatic animal origins were halophilic and grew optionally at 3% NaCl and pH 8.5.

### 3.3. Genome Features of the Six V. parahaemolyticus Isolates Originating in Edible Aquatic Animals

Draft genome sequences of the six *V. parahaemolyticus* isolates were determined using the Illumina Hiseq × 10 (Illumina, San Diego, CA, USA) sequencing platform. Approximately 59,132 to 97,812 clean single reads were obtained. The final assembly yielded 36 to 77 scaffolds with the sequencing depth (on average) of 220.27-fold to 285.8-fold, which varied as a typical Poisson distribution, showing a clean single peak in the frequency of observed unique 17-mers within the sequencing data. No taper at the end of scaffolds or contigs was observed, suggesting less repetitive DNA in the genomes (0.61% to 0.79%) (Figure S1).

The obtained genome sizes ranged from 4,937,042 bp to 5,067,778 bp with GC contents from 45.20% to 45.39%. A total of 4749 to 4919 genes were predicted, including 4622 to 4791 protein-coding genes. Among these, approximately 3852 to 3892 genes were classified

into 22 functional catalogues in the COG database. Remarkably, the unknown function genes (S) accounted for the largest proportion (27.55% to 28.49%) (Table 1, Figure 3).

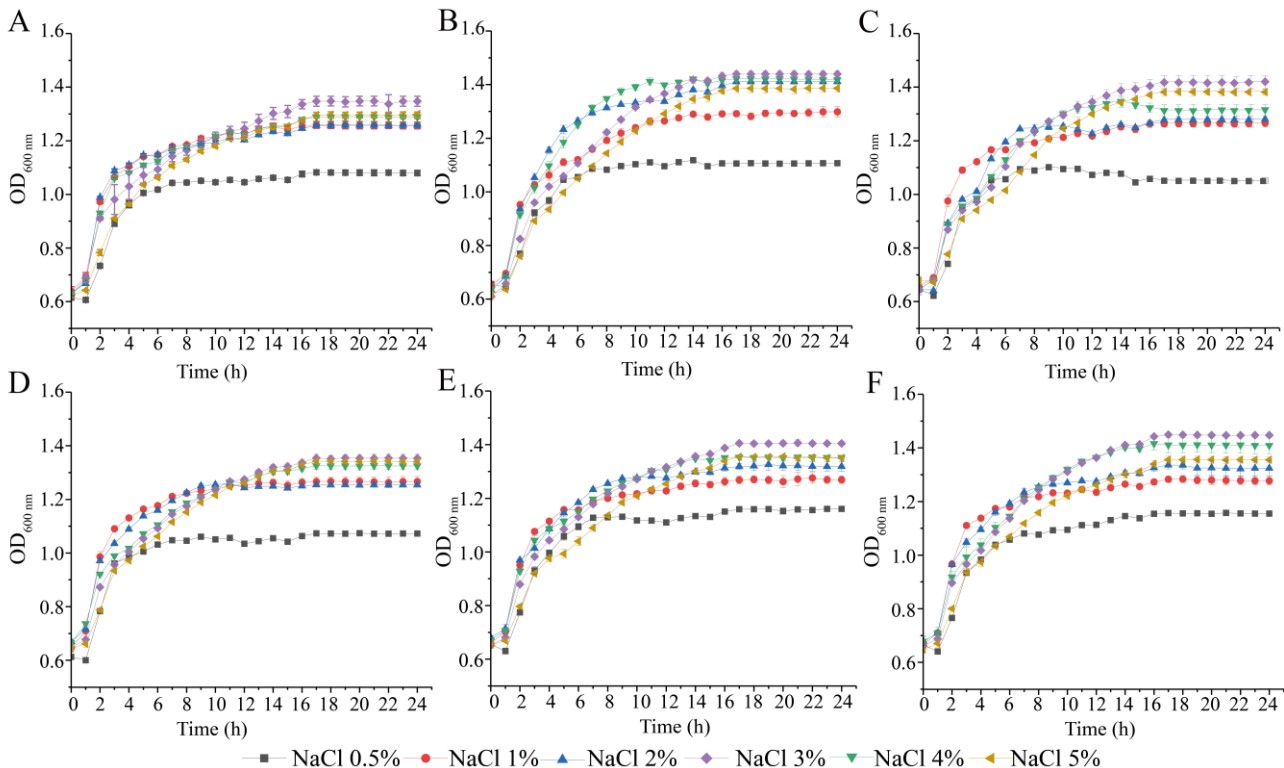

**Figure 2.** Survival of six *V. parahaemolyticus* isolates under different concentrations of NaCl. The isolates were incubated in TSB medium (pH 8.5) at 37 °C. (**A–F**): *V. parahaemolyticus* L7-7, N1-22, N3-33, N4-46, N8-42, and Q8-15, respectively.

The six *V. parahaemolyticus* genomes contained genes for essential cellular metabolisms, such as nucleotide transport and metabolism, energy production and conversion, and carbohydrate transport and metabolism. Most genes involved in information storage and processing were also present in the *V. parahaemolyticus* genomes, including those required for transcription, translation, and ribosomal structure. The genes responsible for cellular processes and signaling (such as defense mechanisms, signal transduction mechanisms, and cell motility) were also identified. Additionally, the six *V. parahaemolyticus* genomes also carried numerous transposase genes (n = 28) and diverse MGEs including GIs (n = 39), prophage gene clusters (n = 5), integrons (n = 33) and ISs (n = 6), suggesting the possibility of HGT mediated by the MGEs during the *V. parahaemolyticus* genome evolution. Notably, only one of these identified MGEs (the IS002 in *V. parahaemolyticus* L7-7 genome) existed at the end of one of the scaffolds (Tables S2–S4), which indicated that the assembled draft genomes did not cause the identified MGEs (except this IS002) to be missing.

The draft genomes of the *V. parahaemolyticus* L7-7, N1-22, N3-33, N4-46, N8-42, and Q8-15 isolates were deposited in GenBank database under the accession numbers JAHRCZ000000000, JAHRDA000000000, JAHRDB000000000, JAHRDC000000000, JAHRDD 000000000, and JAHQCP000000000.

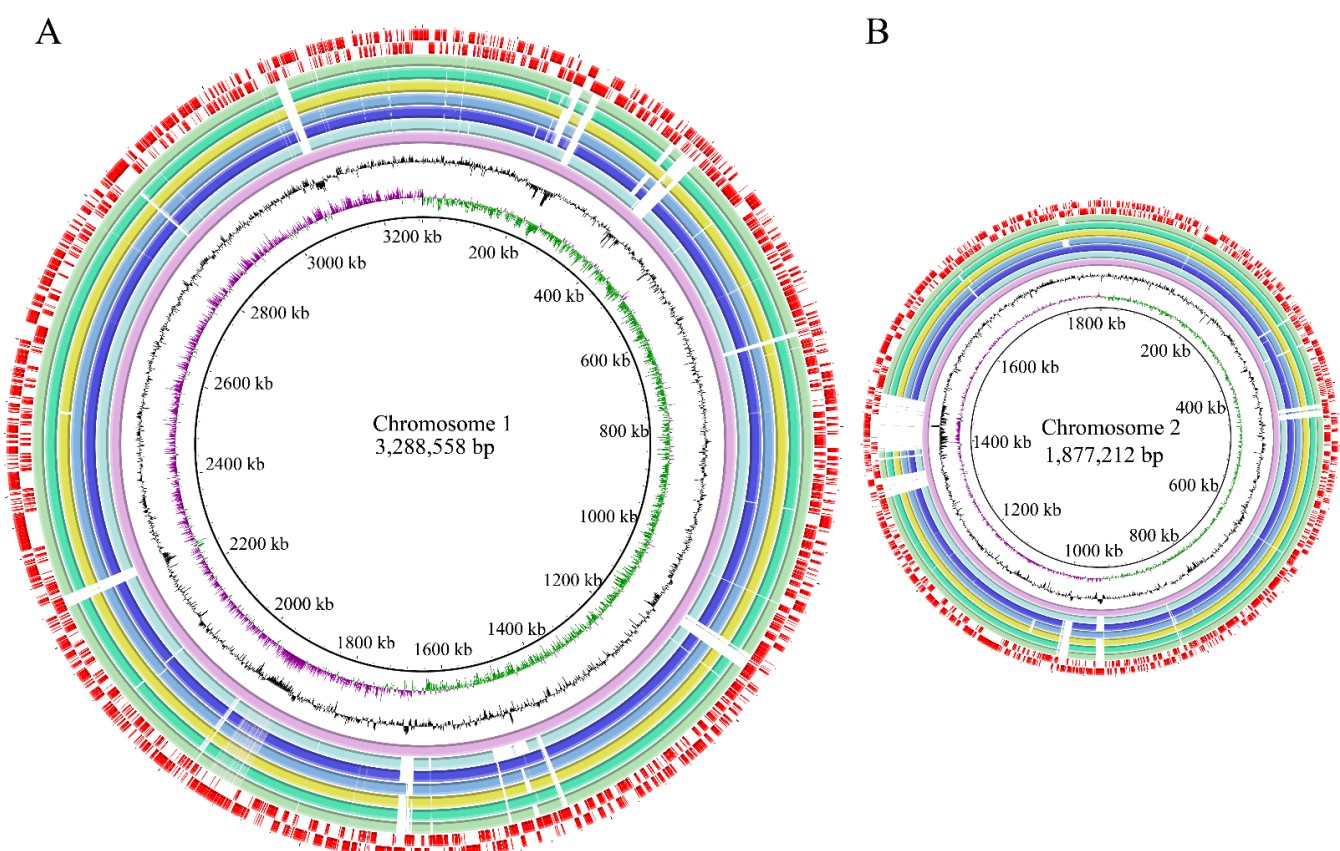

**Figure 3.** Genome circle maps of the six *V. parahaemolyticus* isolates. (**A**,**B**): represents the larger and smaller chromosomes of the six *V. parahaemolyticus* isolates, respectively. Circles from the inwards to outside: the first circle represents GC-skew (purple value is greater than zero, green value is less than zero); the second circle (in black), GC contents (outward part means higher than average, inward part means lower than average); the third circle (in light purple), the reference genome of *V. parahaemolyticus* RIMD 2210633; the fourth to nineth circles (in light blue to light green), *V. parahaemolyticus* L7-7, N1-22, N3-33, N4-46, N8-42, and Q8-15 genomes, respectively; and the tenth circle (in red), CDSs on the positive and negative chains (inward and outward parts), respectively.

### 3.4. Phylogenetic Relatedness of the Six V. parahaemolyticus Isolates Originating in Edible Aquatic Animals

As shown in Figure 4, a phylogenetic tree was construed on the basis of a total of 3447 homologous amino acid sequences identified from 69 *V. parahaemolyticus* genomes analyzed in this study, among which complete genome sequences of 63 *V. parahaemolyticus* genomes were available and retrieved from the GenBank database. Among the engaged 63 *V. parahaemolyticus* isolates, 36 isolates were derived from human, followed by penaeus (n = 14) and homo sapiens (n = 9), while 4 isolates were recovered from the environment from finespotted flounder, shrimp, and toothfish. This analysis revealed seven distinct clusters, designated as Clusters A to G (Figure 4). Among them, *V. parahaemolyticus* N3-33 (GenBank accession no. JAHRDB000000000) recovered from *P. undulate* had the closest evolutionary distance with *V. parahaemolyticus* FDAARGOS_667 (GenBank accession no. NZ_CP044062.1) isolated from homo sapiens, which were classified into Clusters C and B, respectively. Both these strains were phylogenetically closer to *V. parahaemolyticus* L7-7 (GenBank accession no. JAHRCZ000000000) from *A. nobilis*, which fell into a single clade (Cluster A). Similarly, *V. parahaemolyticus* N8-42 (GenBank accession no. JAHRDD000000000) from *M. veneriformis* (Cluster E) was closer to *V. parahaemolyticus* FORC_072 (GenBank accession no. NZ_CP023472.1) originating in human. The latter was grouped into Cluster F together with *V. parahaemolyticus* N4-46 (GenBank

accession no. JAHRDC000000000) from *P. viridis*. Additionally, *V. parahaemolyticus* N1-22 (GenBank accession no. JAHRDA000000000) from *L. vannamei* was classified into a subclade in Cluster G, together with *V. parahaemolyticus* PB1937 (GenBank accession no. NZ_CP022243.1) originating in human. Although *V. parahaemolyticus* Q8-15 (GenBank accession no. JAHQCP000000000) from *C. auratus* was also grouped into Cluster G, it was phylogenetically distant from the other five *V. parahaemolyticus* genomes sequenced in this study. The Q8-15 isolate had closer evolutionary relatedness with *V. parahaemolyticus* 20160303005-1 (GenBank accession no. NZ_CP034298.1) isolated from penaeus (Cluster G) (Figure 4). These results demonstrated genome diversity of the *V. parahaemolyticus* isolates originating in the six species of edible aquatic animals.

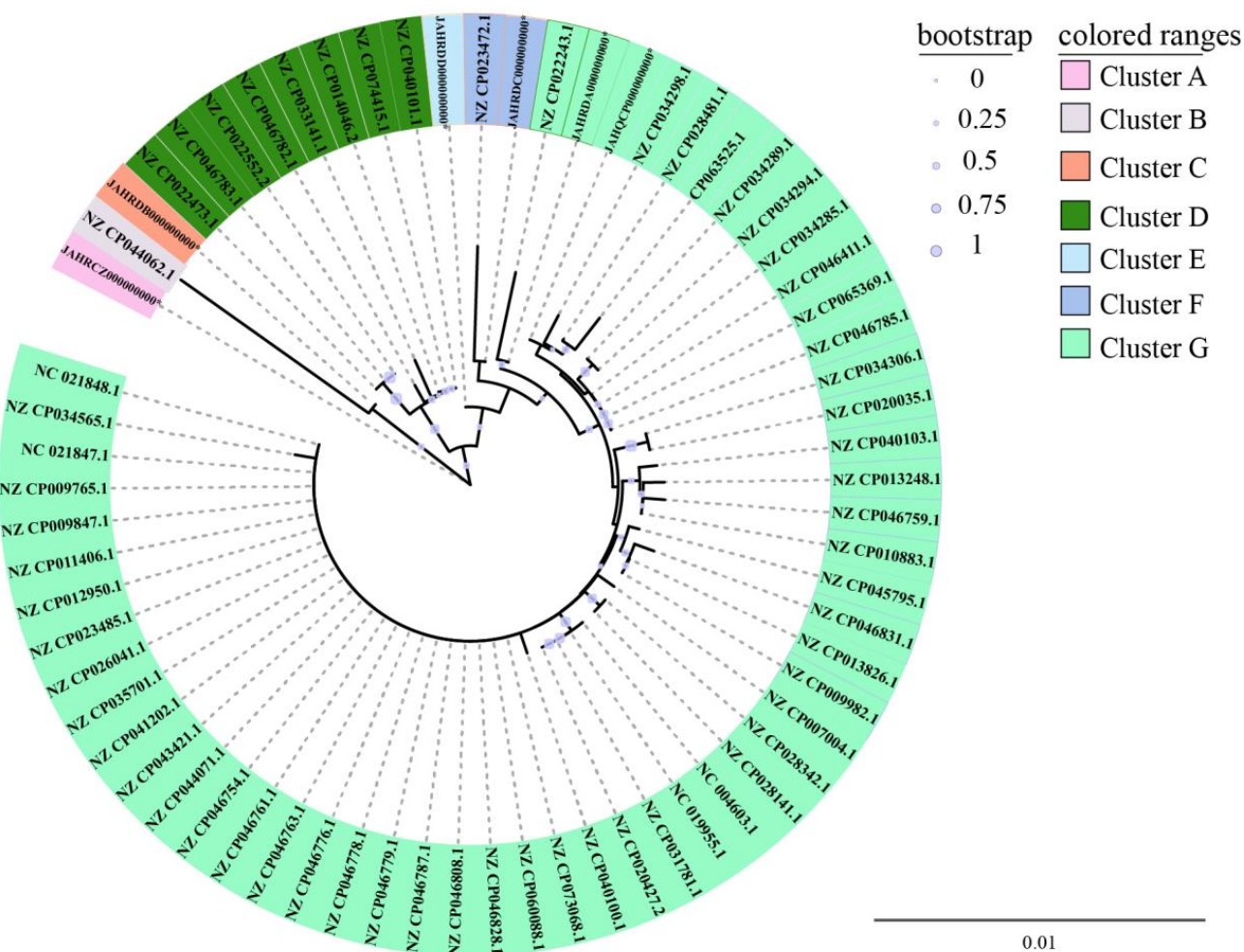

**Figure 4.** Phylogenetic tree showing the relationship of the 69 *V. parahaemolyticus* genomes. Complete genome sequences of the 63 *V. parahaemolyticus* isolates were retrieved from GenBank database with accession numbers showed in the tree. *: genome sequences of the six *V. parahaemolyticus* isolates determined in this study. The phylogenetic tree was constructed using the maximum likelihood approach with 1000 bootstrap replications, and the cutoff threshold for bootstrap values was above 50%.

### 3.5. Diverse MGEs in the Six V. parahaemolyticus Genomes of Edible Aquatic Animal Origins

3.5.1. GIs

GIs can carry large foreign DNA fragments (~200 Kb) that benefit bacterial survival in hosts and in the environment [41,42]. In this study, a total of 39 GIs (Table S2) were identified in the six *V. parahaemolyticus* genomes, which contained 5 to 9 GIs ranging from 4022 bp to 85,856 bp, carrying 3 to 79 genes. Diverse biological functions were observed among the genes carried by the GIs (Figure 5).

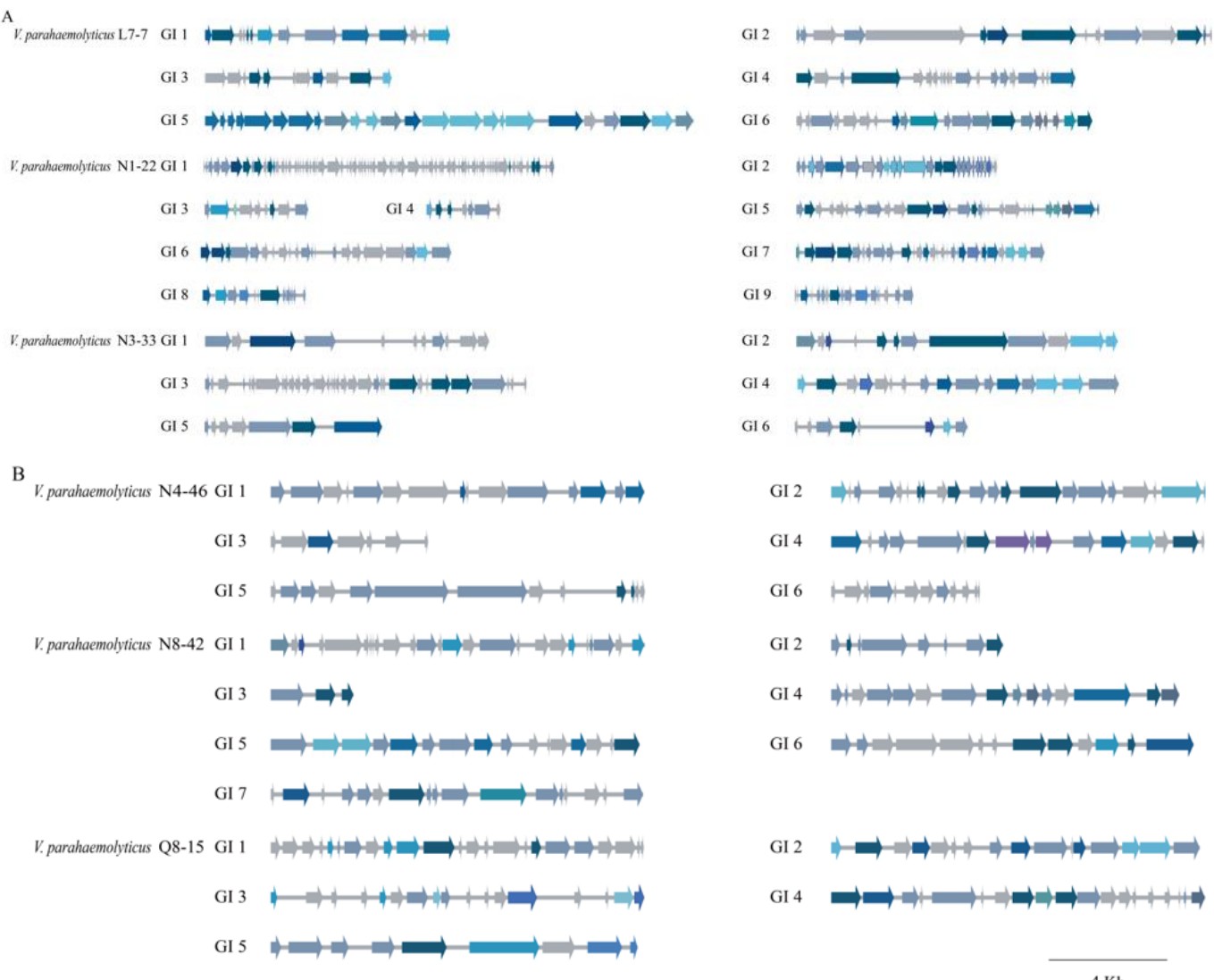

**Figure 5.** Gene organizations of the GIs identified in the six *V. parahaemolyticus* genomes (**A**,**B**). Different colors referred to COG classification to mark gene functions and genes not annotated to COG database were displayed in grey.

For instance, *V. parahaemolyticus* N1-22 originating in *L. vannamei* contained the maximum number of GIs (n = 9, designated as GI 1 to GI 9), which ranged from 9240 bp to 85,856 bp. The largest GI 1 (85,856 bp) carried 79 genes, 18 of which had known functions, including a lipoprotein (*Vp N1-22_0320*), a DUF262 domain-containing protein (*Vp N1-22_0322*), an AAA family ATPase (*Vp N1-22_0323*), a type III restriction-modification system endonuclease (*Vp N1-22_0324*), a site-specific DNA-methyltransferase (*Vp N1-22_0325*), CreA family proteins (*Vp N1-22_0326*, *Vp N1-22_0396*), an ATP-binding protein (*Vp N1-22_0327*), a DUF4113 domain-containing protein (*Vp N1-22_0329*), a S24 family peptidase (*Vp N1-22_0330*), XRE family transcriptional regulators (*Vp N1-22_0363*, *Vp N1-22_0368*), a SAM-dependent DNA methyltransferase (*Vp N1-22_0369*), a DUF285 domain-containing protein (*Vp N1-22_0372*), a DNA adenine methylase (*Vp N1-22_0386*), recombinase family proteins (*Vp N1-22_0394*, *Vp N1-22_0395*), and an inosine/guanosine kinase (*Vp N1-22_0397*). The other genes coded for unknown proteins (n = 61).

Remarkably, there were 9 identified GIs carrying virulence-related genes, including the GI 1 and GI 2 in *V. parahaemolyticus* N1-22 genome; the GI 3 in *V. parahaemolyticus* Q8-15 genome; the GI 3 in *V. parahaemolyticus* L7-7 genome; the GI 4 in *V. parahaemolyticus*

N3-33 genome; the GI 5 and GI 6 in *V. parahaemolyticus* N4-46 genome; and the GI 1 and GI 2 in *V. parahaemolyticus* N8-42 genome. For example, 7 GIs were identified in *V. parahaemolyticus* N8-42 isolated from *M. veneriformis*. Among them, the GI 2 (31,937 bp) was the longest and carried 28 genes, 10 of which have known functions, including a type III toxin-antitoxin system ToxN/AbiQ family toxin (*Vp N8-42_1768*), a helix-turn-helix domain-containing protein (*Vp N8-42_1769*), a replication protein P (*Vp N8-42_1775*), a type IV secretion system (T4SS) DNA-binding domain-containing protein (*Vp N8-42_1779*), a relaxase domain-containing protein (*Vp N8-42_1780*), IS5 family transposases (*Vp N8-42_1785, Vp N8-42_1791*), an AAA family ATPase (*Vp N8-42_1792*), a site-specific integrase (*Vp N8-42_1794*), and an xYicC family protein (*Vp N8-42_1795*). The other genes coded for unknown proteins (n = 18). Interestingly, the GI 3, and GI 4 in *V. parahaemolyticus* Q8-15, and N8-42 genomes contained the same gene (*Vp Q8-15_1623; Vp N8-42_2908*) encoding a type II toxin-antitoxin system RelE/ParE family toxin. The GI 7, GI 2, GI 3, and GI 4 in *V. parahaemolyticus* N1-22, Q8-15, L7-7, and N3-33 genomes carried the same *SmpB* gene (*Vp N1-22_3937; Vp Q8-15_2778; Vp L7-7_3357; Vp N3-33_2957*), which encoded an SsrA-binding protein, SmpB, that was crucial to the virulence expression of pathogenic *Aeromonas veronii* [43].

Additionally, the genes encoding hydrolysis enzymes, chemotaxis proteins, stress regulators, and resistance-related proteins were also identified in some GIs in the six *V. parahaemolyticus* genomes. For example, there were two identified GIs carrying heavy-metal-tolerance-related genes: specifically, the GI 6 and GI 4 in the *V. parahaemolyticus* L7-7 and N8-42 genomes, respectively. Of these, the gene encoding a zinc uptake transcriptional repressor Zur (*Vp L7-7_4298*) was identified in the GI 6 of *V. parahaemolyticus* L7-7, which was an important regulator of bacterial zinc metabolism [44].

Interestingly, there were several GIs carrying phage regulator genes, such as the GI 2 and GI 6 in *V. parahaemolyticus* N1-22; the GI 3 in *V. parahaemolyticus* L7-7; the GI 3 in *V. parahaemolyticus* N3-33; the GI 3 and GI 6 in *V. parahaemolyticus* N4-46; and the GI 5 in *V. parahaemolyticus* N8-42. For example, the GI 2 in *V. parahaemolyticus* N1-22 genome contained the *Vp N1-22_1312* and *Vp N1-22_1310* genes encoding a phage regulatory CII family protein and a phage repressor protein CI, respectively, while the GI 6 contained the *Vp N1-22_3646* gene encoding a phage integrase family protein. Similarly, the gene (*Vp L7-7_3352; Vp N4-46_3330; Vp N8-42_3479*) encoding an AlpA family phage regulatory protein was identified in the GI 3, GI 3, and GI 5 in *V. parahaemolyticus* L7-7, N4-46, and N8-42 genomes, respectively.

3.5.2. Prophages

Prophages are closely related to bacterial pathogenicity, especially via the transfer of virulence factors [45]. In this study, five prophage gene clusters were identified in *V. parahaemolyticus* N1-22, N4-46, N8-42, and Q8-15 genomes (Table S3), but absent from *V. parahaemolyticus* L7-7, and N3-33 genomes. The latter two genomes also contained several phage regulator genes, implying possible phage DNA integration during their genome evolution. The identified prophage gene clusters ranged from 10,131 bp to 32,968 bp and encompassed 13 to 40 genes (Figure 6). *V. parahaemolyticus* N8-42 genome contained two prophage gene clusters with sequence similarity to *Vibrio*_phage_K139 (33,106 bp, NCBI accession number: NC_003313) and *Vibrio*_phage_fs2 (8651 bp, NCBI accession number: NC_001956), respectively. *V. parahaemolyticus* N4-46, Q8-15, and N1-22 genomes contained one homologous prophage similar to *Pseudomonas*_phage_D3 (56,426 bp, NCBI accession number: NC_002484), *Pseudomonas*_phage_D3, and *Vibrio*_phage_K139 (33,106 bp, NCBI accession number: NC_003313), respectively.

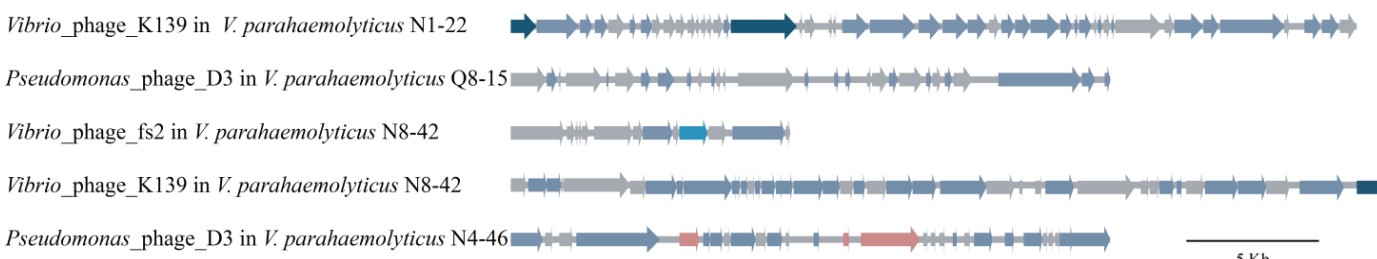

**Figure 6.** Structure diagram of the prophage gene clusters identified in the *V. parahaemolyticus* genomes.

For example, in *V. parahaemolyticus* N8-42 genome, the identified *Vibrio*_phage_K139 (32,968 bp) encoded 40 genes, including eight phage-related genes such as phage tail proteins (*Vp N8-42_2875*, *Vp N8-42_2889*), a baseplate J/gp47 family protein (*Vp N8-42_2877*), a phage tail tape measure protein (*Vp N8-42_2879*), a head completion/stabilization protein (*Vp N8-42_2890*), a phage major capsid protein and P2 family protein (*Vp N8-42_2892*), a phage portal protein (*Vp N8-42_2895*), and a phage regulatory CII family protein (*Vp N8-42_2903*). There were 15 genes encoding functional proteins, e.g., a hemolysin (*Vp N8-42_2876*), a DUF2590 family protein (*Vp N8-42_2878*), a TraR/DksA C4-type zinc finger protein (*Vp N8-42_2885*), a DUF2597 family protein (*Vp N8-42_2886*), a DUF2586 family protein (*Vp N8-42_2887*), terminases (*Vp N8-42_2891*, *Vp N8-42_2894*), a GPO family capsid scaffolding protein (*Vp N8-42_2893*), an ogr/Delta-like zinc finger family protein (*Vp N8-42_2896*), a helix-turn-helix domain-containing protein (*Vp N8-42_2905*), a DUF262 domain-containing protein (*Vp N8-42_2906*), a DUF3696 domain-containing protein (*Vp N8-42_2907*), a type II toxin-antitoxin system RelE/ParE family toxin (*Vp N8-42_2908*), a peptidoglycan DD-metalloendopeptidase family protein (*Vp N8-42_2909*), and a tyrosine-type recombinase/integrase (*Vp N8-42_2910*). Additionally, 17 genes encoded unknown proteins. Our results provided additional evidence for the existence of *Vibrio*_phage_K139 homologues in *V. parahaemolyticus* isolates [46].

The *V. parahaemolyticus* N8-42 genome also contained *Vibrio*_phage_fs2 (10,537 bp), the shortest prophage identified in this study. The identified *Vibrio*_phage_fs2 contained 13 genes, 7 of which coded for known proteins, e.g., bacterial type II and III secretion (T2SS and T3SS) family protein (*gspD*) (*Vp N8-42_0752*). The conserved GspD secreted by T2SS was a putative functionally important protein in pathogenic *Leptospira interrogans* [47].

The *Pseudomonas*_phage_D3 homologues (22,614 bp; 22,820 bp) were identified in *V. parahaemolyticus* N4-46, and Q8-15 genomes, respectively. They carried 28 and 29 genes, 10 of which coded for known proteins. Of these, although phage major structure genes were missing, the *cspA* gene (*Vp N4-46_4569*; *Vp Q8-15_2045*) encoding a cold-shock protein was identified, which played a crucial role in adapting to the adverse environment—including environmental behaviors of antibiotics in Himalayan psychrotolerant *Pseudomonas* strains [48]. Notably, the presence of *Pseudomonas*_phage_D3 homologues in *V. parahaemolyticus* N4-46, and Q8-15 genomes suggested possible HGT across different genera of *Pseudomonas* and *Vibrio*.

Additionally, the *Vibrio*_phage_K139 homologue (31,909 bp) was also present in the *V. parahaemolyticus* N1-22 genome. It carried 46 genes, including the phage major structure genes, which were the same as those identified in the *V. parahaemolyticus* N8-42 genome. However, seven different accessory genes were identified, encoding a DUF4041 domain-containing protein (*VpN1_22_1307*), a PH domain-containing protein (*VpN1_22_1308*), a phage repressor protein CI (*VpN1_22_1310*), a replication endonuclease (*VpN1_22_1322*), an RNA-binding protein (*VpN1_22_1321*), a tail assembly chaperone (*VpN1_22_1342*), and a virion morphosis protein (*VpN1_22_1334*). These results suggested that more recent HGT and genome recombination likely occurred among the *V. parahaemolyticus* isolates.

3.5.3. Integrons

Mobile integrons were prevalent in human-dominated ecosystems with prolonged exposure to detergents, antibiotics, and heavy metals [49]. Integrons are generally classified

according to integrase genes (*intI1*, *intI2*, *intI3*, and *intI4*) into type I, type II, type III, and super integron, respectively [50,51]. In this study, all six *V. parahaemolyticus* genomes contained integrons (n = 1 to 11) ranging from 1082 bp to 39,523 bp (Figure 7).

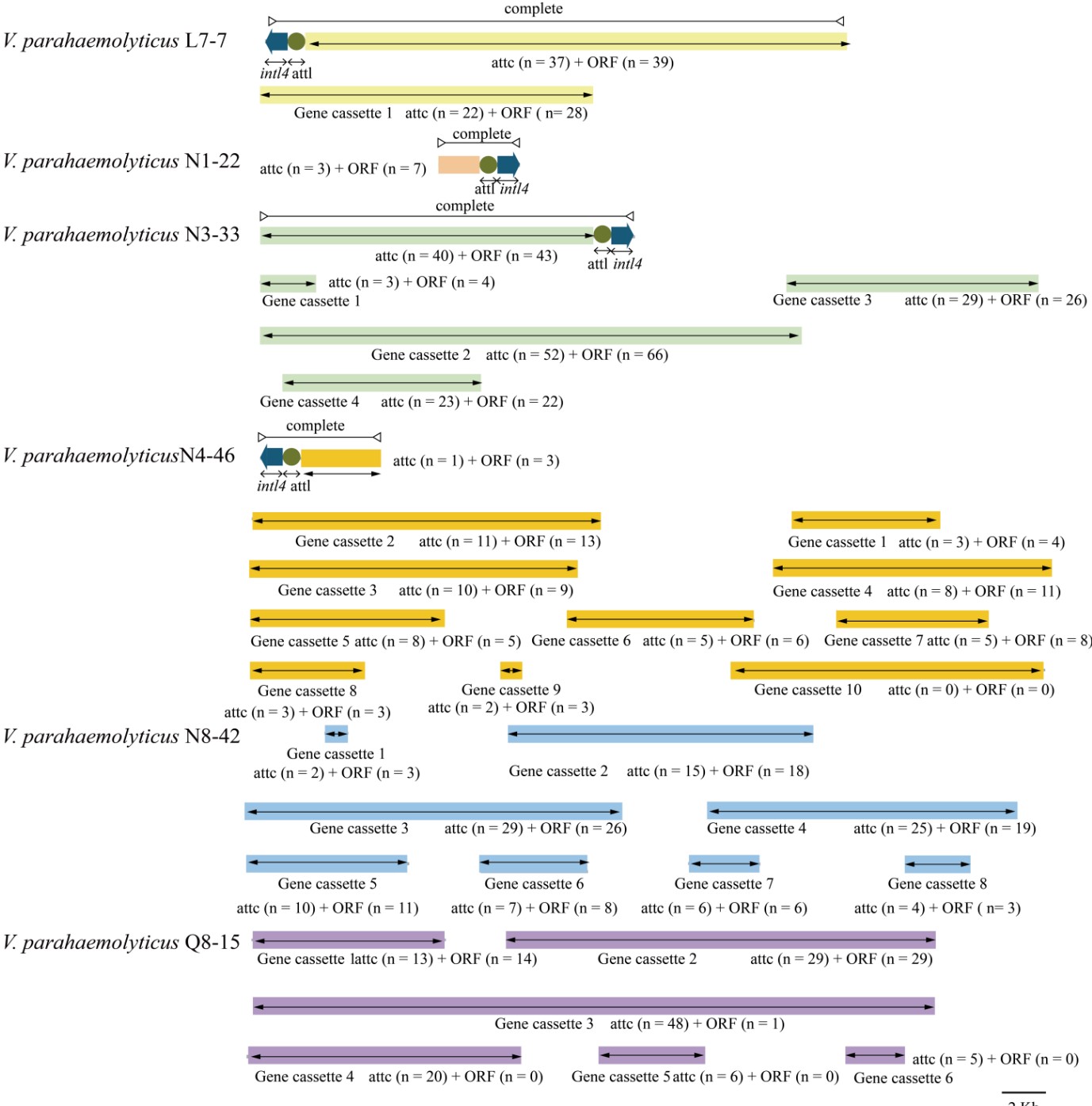

**Figure 7.** Structure diagram of the integrons identified in the six *V. parahaemolyticus* genomes. The complete integrons and incomplete gene cassettes identified in *V. parahaemolyticus* L7-7, N1-22, N3-33, N4-46, N8-42, and Q8-15 genomes are shown from light yellow to light purple, with the predicted *attc/attl* sites and ORFs.

Remarkably, *V. parahaemolyticus* N4-46 isolated from *P. viridis* contained the maximum number of integrons (n = 11) ranging from 1082 bp to 14,532 bp. The 11 identified integrons (designated as In 1 to In 11) were divided into two categories: complete integron (n = 1)

and gene cassettes (n = 10). For example, the ln 2 (1919 bp) contained four genes, encoding a flagellin *(Vp N4-46_*2269), hypothetical proteins *(Vp N4-46_*2648, *Vp N4-46_*2649) and a site-specific recombinase Intl4 *(Vp N4-46_*2647), indicating that it was a super integron.

Only one integron was identified in the *V. parahaemolyticus* N1-22 genome. It was a complete integron (3738 bp) containing 7 genes, encoding a type II toxin-antitoxin system RelB/DinJ family antitoxin (*Vp N1-22_*3060), a type II toxin-antitoxin system RelE/StbE family mRNA interferase toxin (*Vp N1-22_*3061), a YkgJ family cysteine cluster protein (*Vp N1-22_*3062), a type II toxin-antitoxin system Phd/YefM family antitoxin (*Vp N1-22_*3063), a type II toxin-antitoxin system RelE/ParE family toxin (*Vp N1-22_*3064), a conserved hypothetical protein (*Vp N1-22_*3065), and an integrase Intl4 (*Vp N1-22_*3066), indicating that it was also a super integron. The type II toxin-antitoxin system was initially discovered in plasmids, showing a phenomenon of post-segregational killing. They were later shown to be present in bacterial chromosomes, often in association with MGEs [52].

The *V. parahaemolyticus* L7-7 genome contained 2 integrons: a complete integron and an incomplete integron. The former, In 1 (26,693 bp), was a super integron and contained 40 genes, 11 of which have known functions, encoding a NAD(P)H-dependent oxidoreductase (*Vp L7-7_*3000), a DUF4238 domain-containing protein (*Vp L7-7_*3002), a DUF2569 domain-containing protein (*Vp L7-7_*3004), a DUF4062 domain-containing protein (*Vp L7-7_*3013), a GIY-YIG nuclease family protein (*Vp L7-7_*3014), GNAT family N-acetyltransferases (*Vp L7-7_*3015, *Vp L7-7_*3020), a putative NADPH-P-450 reductase (*Vp L7-7_*3016), a glutathione S-transferase (*Vp L7-7_*3026), a nucleotide pyrophosphohydrolase (*Vp L7-7_*3031), and an integrase Intl4 (*Vp L7-7_*2993).

In the *V. parahaemolyticus* N3-33 genome, 5 integrons were identified, including one complete In 1 and 4 incomplete integrons (In 2 to In 5). The In 1 (27,052 bp) contained 44 genes, 12 of which have known functions, encoding DUF3265 domain-containing proteins (*Vp N3-33_*1753, *Vp N3-33_*1762, *Vp N3-33_*1787), an effector-binding domain-containing protein (*Vp N3-33_*1754), a NUDIX domain-containing protein (*Vp N3-33_*1761), GNAT family N-acetyltransferases (*Vp N3-33_*1764, *Vp N3-33_*1774, *Vp N3-33_*1777), an NAD-dependent DNA ligase (*Vp N3-33_*1766), a DUF4145 domain-containing protein (*Vp N3-33_*1773), a histone acetyltransferase HPA2-related acetyltransferase (*Vp N3-33_*1779), a methyltransferase domain-containing protein (*Vp N3-33_*1784), and an integrase Intl4 (*Vp N3-33_*1796), indicating that this integron was also a super integron.

Interestingly, the identified four complete integrons in *V. parahaemolyticus* N4-46, N1-22, L7-7, and N3-33 genomes were all super integrons, one of which in the *V. parahaemolyticus* N1-22 genome carried virulence-related genes. The super integron was a common feature among the *Vibrio* species, but it was also highly variable [53]. Additionally, eight and six incomplete integrons were identified in the *V. parahaemolyticus* N8-42 and Q8-15 genomes, respectively.

### 3.5.4. ISs

ISs are the shortest autonomously mobile elements (<2.5 Kb) that have simple genetic organization but can insert at multiple sites in a target molecule [54]. In this study, ISs were identified in *V. parahaemolyticus* L7-7 (n = 3), N3-33 (n = 1), N8-42 (n = 1), and Q8-15 (n = 1) genomes (Table S4) but were absent from *V. parahaemolyticus* N1-22 and N4-46 genomes.

The *V. parahaemolyticus* L7-7 genome had 3 ISs: IS001 with 1232 bp that contained a mobile element protein (*Vp L7-7_*4694) and a IS3 family transposase (*Vp L7-7_*4695); IS002 with 1053 bp carrying a IS5 family transposase (*Vp L7-7_*4745); and IS003 with 869 bp carrying a AraC family transcriptional regulator (*Vp L7-7_*2087).

The IS (1029 bp) belonging to the IS3 family was also present in the *V. parahaemolyticus* N3-33 genome, encoding hypothetical proteins (*Vp N3-33_*3401, *Vp N3-33_*3402), while that (962 bp) belonging to the IS5 family also existed in *V. parahaemolyticus* Q8-15 genome, carrying a GyrI-like domain-containing protein (*Vp Q8-15_*0248). These results suggested that these ISs were possibly capable of skipping in these *V. parahaemolyticus* genomes.

Additionally, one IS (1419 bp) belonging to the IS91 family was identified in *V. parahaemolyticus* N8-42 genome, within which no gene located.

### 3.5.5. ICEs

The BLAST searching of the specific *prfC* gene (Genbank Accession No: NC_004603.1) of *V. parahaemolyticus,* and the genes encoding conserved module structures of SXT/R391 ICEs, showed that ICE was absent from the six *V. parahaemolyticus* genomes analyzed in this study.

### 3.6. CRISPR-Cas Systems

CRISPR-Cas systems are prokaryotic immune systems that can recognize foreign DNA and resist foreign genetic material or genes, which are found in about 84% of archaea and 47% of bacterial genomes [55]. In this study, different numbers of CRISPR-Cas gene clusters (97 bp to 825 bp) were identified in *V. parahaemolyticus* N3-33 (n = 3), N1-22 (n = 2), and Q8-15 (n = 1) genomes, but none of which contained the Cas protein-encoding gene, suggesting incomplete CRISPR-Cas systems in the 3 *V. parahaemolyticus* isolates (Figure 8).

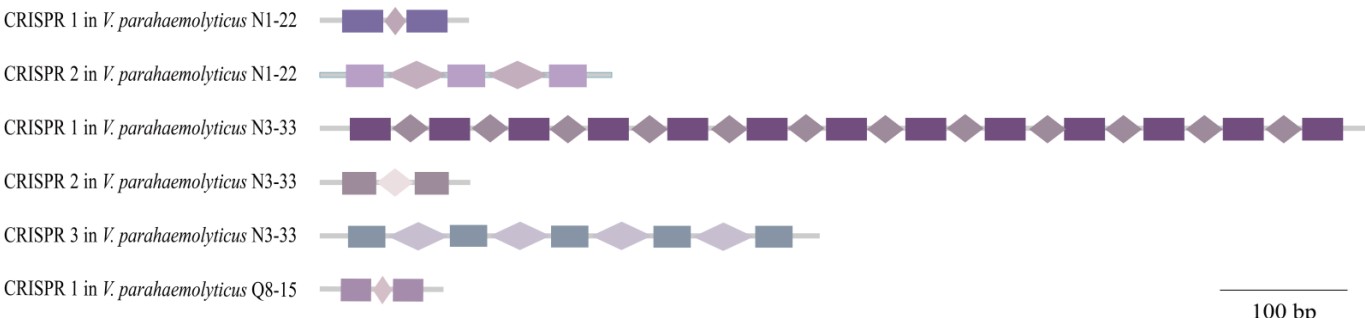

**Figure 8.** The structural features of CRISPRs identified in the *V. parahaemolyticus* genomes. The repeat sequences were shown by the rectangle of different colors and the spacer regions were represented by rhombuses of different colors.

For instance, *V. parahaemolyticus* N3-33 had 3 CRISPR-Cas gene clusters, designated as CRISPR1, CRISPR2, and CRISPR3. The CRISPR1 was 825 bp with 13 repeats and 12 spacers; the CRISPR2 was 118 bp with 2 repeats and 1 spacer; and the CRISPR3 was 392 bp with 5 repeats and 4 spacers.

*V. parahaemolyticus* N1-22 genome had two identified CRISPR-Cas clusters: the CRISPR1 (117 bp) had 2 repeats (46 bp) and 1 spacer (26 bp), while the CRISPR2 (229 bp) had 3 repeats (36 bp) and 2 spacers (61 bp).

Only one CRISPR-Cas cluster (97 bp) was identified in *V. parahaemolyticus* Q8-15 genome, consisting of two repeats (36 bp) and a spacer (26 bp). Due to its extremely short length (97 bp) and the fewer repeats (n = 2), this sequence might not have the function of the CRISPR-Cas system.

### 3.7. Putative Virulence-Associated Genes in the Six V. Parahaemolyticus Genomes

The BLAST search for virulence genes in the six *V. parahaemolyticus* genomes revealed distinct virulence gene profiles. Approximately 77 to 79 virulence-related genes were identified in the six *V. parahaemolyticus* genomes. *V. parahaemolyticus* Q8-15 isolate recovered from *C. auratus* contained the maximum number of virulence-associated genes (n = 79), whereas *V. parahaemolyticus* L7-7 had relatively fewer (n = 77).

The *flrBC* and *yscOPQRST* gene clusters were identified in all *V. parahaemolyticus* genomes. The former contributed to virulence of pathogenic *Vibrio* via adhesion or biofilm formation, while the *ysc* cluster was closely related to T3SS [56]. Moreover, the *fliCDE* gene cluster was also identified in the six *V. parahaemolyticus* genomes, which was involved in bacterial virulence through the flagellar mobility. This gene cluster was also found in most *Salmonella* strains [57] and was widespread in *Vibrio cholerae* isolates [36].

The other virulence-associated genes involved in bacterial adhesion, colonization invading, and persisting in the host were also identified in the six *V. parahaemolyticus*

genomes. For example, the genes *impI*, *vgrG*, and *acfA* encoding accessory colonization factors in pathogenic *V. cholerae* and *Klebsiella pneumoniae* [36,58] were identified in the six *V. parahaemolyticus* genomes. A *gspN* gene encoding a T2SS-related protein was found in *V. parahaemolyticus* N1-22 and N3-33 genomes, while a *yscH* gene encoding a T3SS polymerization control protein VscH was identified in all six genomes. A *gmd* gene encoding a GDP-mannose 4,6-dehydratase was found in *V. parahaemolyticus* N4-46, L7-7, and Q8-15 genomes, which was required for the synthesis of colanic acid capsule that indirectly controlled the virulence of bacteria [59]. A *tuf* gene encoding an elongation factor EF-Tu was identified in the *V. parahaemolyticus* Q8-15 genome, which functioned in peptide chain formation to enhance bacterial pathogenicity [60]. Moreover, a *hap* gene encoding a virulent metalloproteinase was found in the six *V. parahaemolyticus* genomes, which directly affected the effect of epithelial tight junction integrity of *V. parahaemolyticus* [61]. Additionally, putative virulence-associated genes were also identified in the six *V. parahaemolyticus* genomes, e.g., flagellar biosynthesis (*flgBM*, *fliQ*, and *flhABF*), chemotaxis (*cheA*), motility (*pilT*), and regulation (*glnGL*) genes. The identified virulence-associated genes could be candidate targets for the development of new diagnostic methods, vaccines, and treatments for controlling *V. parahaemolyticus* infections in humans.

### 3.8. Antibiotic Resistance-Associated Genes in the Six V. parahaemolyticus Genomes

Approximately 17 to 20 antimicrobial resistance-related genes were identified in the six *V. parahaemolyticus* genomes (Table 2). *V. parahaemolyticus* N1-22 and N3-33 isolated from *L. vannamei* and *P. undulate*, respectively, contained the maximum number of resistance genes (n = 20), followed by L7-7, N8-42 and Q8-15 isolates from *A. nobilis*, *M. veneriformis*, and *C. auratus*, respectively (n = 18); and N4-46 isolate from *P. viridis* (n = 17).

All six *V. parahaemolyticus* genomes contained the genes encoding a multidrug efflux RND transporter periplasmic adaptor (*acrB*) [62], a multidrug resistance (MDR) protein (*norM*) [63], a penicillin-binding protein (*mrcB*) [64], and a β-lactamase (*blaCARB-17*). The genes for resistance to TET (*tet34*, *tet35*) and AMP (*crp*) were also identified, consistent with antibiotic-resistant phenotypes of the six *V. parahaemolyticus* isolates. Moreover, the *catB*, *SoxR*, *hns*, *qnr*, *folA*, *macB*, and *msbA* genes were found in the six *V. parahaemolyticus* genomes, which encoded a sugar O-acetyltransferase, a redox-sensitive transcriptional activator SoxR, a DNA-binding protein H-NS, a Qnr family pentapeptide repeat protein, a type 3 dihydrofolate reductase, an MacB family efflux pump subunit, and a lipid A ABC transporter ATP-binding protein/permease MsbA, respectively. They were responsible for resistance to phenicol (*catB*, *SoxR*), fluoroquinolone (*hns*, *qnr*, *crp*), macrolide (*macB*), and nitroimidazole (*msbA*). For example, ABC transporters are transmembrane molecular pumps, which share a typical architectural organization comprising highly conserved hydrophilic nucleotide-binding domains (NBDs) and less conserved hydrophobic transmembrane domains (TMDs). NBDs, also known as ATP-binding cassettes, are located at the cytoplasmic side of the cell membrane and hydrolyze ATP to generate translocation driving energy, while TMDs form the translocation pathway and provide specificity to a substrate [65]. Recently, Kumar et al. analyzed various genetic features of putative ABC transporters classified into A-H subfamilies and delineated their role during mosquito *Aedes aegypti* development and arboviral infection via transcriptome analyses [66]. Additionally, different numbers of *bla* genes encoding beta-lactamases were found in *V. parahaemolyticus* N1-22, N3-33, and N4-46 genomes (Table 2).

**Table 2.** The antimicrobial and heavy metal resistance-associated genes identified in the six *V. parahaemolyticus* genomes.

| Antimicrobial Agent | Gene | *V. parahaemolyticus* Isolates |
|---|---|---|
| Cephalosporin | *acrB* | L7-7, N1-22, N3-33, N4-46, N8-42, Q8-15 |
| Phenicol | *catB* | L7-7, N1-22, N3-33, N4-46, N8-42, Q8-15 |
| | *SoxR* | L7-7, N1-22, N3-33, N4-46, N8-42, Q8-15 |
| Fluoroquinolone | *hns* | L7-7, N1-22, N3-33, N4-46, N8-42, Q8-15 |
| | *acrB* | L7-7, N1-22, N3-33, N4-46, N8-42, Q8-15 |
| | *qnr* | L7-7, N1-22, N3-33, N4-46, N8-42, Q8-15 |
| | *crp* | L7-7, N1-22, N3-33, N4-46, N8-42, Q8-15 |
| Tetracycline | *tet(35)* | L7-7, N1-22, N3-33, N4-46, N8-42, Q8-15 |
| | *tet(34)* | L7-7, N1-22, N3-33, N4-46, N8-42, Q8-15 |
| | *hns* | L7-7, N1-22, N3-33, N4-46, N8-42, Q8-15 |
| | *acrB* | L7-7, N1-22, N3-33, N4-46, N8-42, Q8-15 |
| | *SoxR* | L7-7, N1-22, N3-33, N4-46, N8-42, Q8-15 |
| Beta-lactam | *bla*$_{CARB-18}$ | N3-33 |
| | *bla*$_{CARB-19}$ | L7-7, N1-22, N8-42, Q8-15 |
| | *bla*$_{CARB-21}$ | L7-7, N1-22, N3-33, N8-42, Q8-15 |
| | *bla*$_{CARB-23}$ | N4-46 |
| | *bla*$_{CARB-29}$ | N3-33, N4-46,Q8-15 |
| | *bla*$_{CARB-35}$ | N1-22 |
| | *bla*$_{CARB-33}$ | N3-33 |
| | *bla*$_{CARB-41}$ | N3-33 |
| | *bla*$_{CARB-44}$ | L7-7, N8-42 |
| | *bla*$_{CARB-45}$ | N1-22 |
| | *bla*$_{CARB-48}$ | N1-22 |
| Diaminopyrimidine | *folA* | L7-7, N1-22, N3-33, N4-46, N8-42, Q8-15 |
| Macrolide | *macB* | L7-7, N1-22, N3-33, N4-46, N8-42, Q8-15 |
| Nitroimidazole | *msbA* | L7-7, N1-22, N3-33, N4-46, N8-42, Q8-15 |
| Heavy metal | *cusA* | L7-7, N1-22, N3-33, N4-46, N8-42, Q8-15 |
| Heavy metal | *cusR* | L7-7, N1-22, N3-33, N4-46, N8-42, Q8-15 |
| Heavy metal | *cusS* | L7-7, N1-22, N3-33, N4-46, N8-42, Q8-15 |
| Heavy metal | *zntA* | L7-7, N1-22, N3-33, N4-46, N8-42, Q8-15 |
| Heavy metal | *copA* | L7-7, Q8-15 |

*3.9. Heavy Metal Tolerance-Associated Genes in the Six V. parahaemolyticus Genomes*

In this study, approximately five heavy metal tolerance-associated genes were identified in the six *V. parahaemolyticus* genomes (Table 2). For example, the *cusARS* and *zntA* genes were identified in all six *V. parahaemolyticus* genomes, which encoded heavy metal efflux RND transporter and Zn/Cd/Hg/Pb-transporting ATPase, respectively. The *copA* gene encoding heavy metal translocating P-type ATPase was also identified in *V. parahaemolyticus* L7-7 and Q8-15 genomes (Table 2).

*3.10. Strain-Specific Genes of the Six V. parahaemolyticus Isolates of Edible Aquatic Animal Origins*

Comparative genomic analyses revealed approximately 4081 core genes, which accounted for a large proportion (75.7%) of pan genes (n = 5392) and were conserved among the six *V. parahaemolyticus* genomes. Meanwhile, comparative genomic analyses revealed a number of strain-specific genes (n = 131 to 287) in the *V. parahaemolyticus* isolates (Figure 9). Interestingly, *V. parahaemolyticus* N3-33 isolate originating in *P. undulat* contained the maximum number of strain-specific genes (n = 287), whereas *V. parahaemolyticus* N8-42 isolate in *M. veneriformis* had the minimum (n = 131). Remarkably, higher percentages of strain-specific genes (41.0% to 60.3%) encoded unknown proteins, while most of the others were involved in outer membrane biogenesis, cell envelope, and secondary metabolisms. These results provided additional evidence for the genome diversity of the six *V. parahaemolyticus* isolates of edible aquatic animal origins.

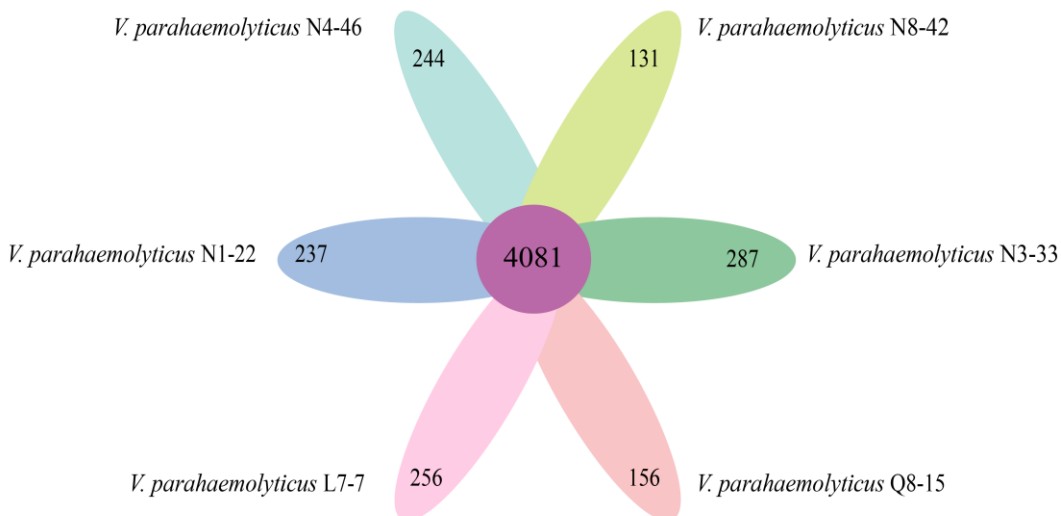

**Figure 9.** Venn diagram showing the core genes and strain-specific genes in the six *V. parahaemolyticus* genomes.

## 4. Discussion

*V. parahaemolyticus* has consistently caused foodborne disease outbreaks every year worldwide [67]. Continuous monitoring of *V. parahaemolyticus* contamination in aquatic products is imperative for food safety systems, particularly in developing nations. In the present study, we characterized genome features of *V. parahaemolyticus* isolates isolated from 6 species of edible aquatic animals, including *A. nobilis*, *L. vannamei*, *P. undulate*, *P. viridis*, *M. veneriformis*, and *C. auratus*. Our data showed that most *V. parahaemolyticus* isolates grew optimally in TSB medium at pH 8.5 and 3% NaCl, consistent with previous report [23]. Diverse antibiotic and heavy metal resistance profiles derived from the six *V. parahaemolyticus* isolates suggested different antibiotic and heavy metal exposure levels or pollution sources in aquaculture environments.

In this study, draft genomes of the six *V. parahaemolyticus* isolates were determined (4,937,042 bp to 5,067,778 bp). Some repeats were observed at the end of scaffolds (n = 2 to 6, <1.3 Kb) in five *V. parahaemolyticus* genomes (Table S5). Therefore, it is reasonable to speculate that the unassembled gaps between scaffolds were comprised of repetitive DNA. Approximately 4622 to 4791 protein-coding genes were annotated, of which 1064 to 1107 encoded unknown proteins. Moreover, comparative genomic analyses revealed a number of strain-specific genes (n = 131 to 287) in the six *V. parahaemolyticus* isolates, approximately 41.0% to 60.3% of which encoded unknown proteins. Various MGEs were also identified, including GIs (n = 5 to 9), prophage gene clusters (n = 0 to 2), CRISPR-Cas systems (n = 0 to 6), and integrons (n = 1 to 11), and ISs (n = 0 to 3). Of course, we could not rule out the possibility that additional genetic elements could be present in the gaps between scaffolds. These results demonstrated genome variation among the six *V. parahaemolyticus* isolates originating in edible aquatic animals.

Intercellular transmissibility of the identified MGEs carrying many genes may have constituted important driving forces in *V. parahaemolyticus* genome evolution and speciation. For instance, the identified 39 GIs (4022 bp to 85,856 bp) in the six *V. parahaemolyticus* genomes contained 712 genes, which endowed the bacterium with diverse biological functions, such as virulence, second metabolisms, stress response, and resistance. Of these, nine GIs carrying virulence-related genes were identified. Moreover, there were several GIs containing phage regulator genes. These results provided additional evidence for the presence of GIs closely related to enteropathogenicity in *V. parahaemolyticus* [68]. Pazhani et al. reported that acquisition of GIs by HGT provides enhanced tolerance of *V. parahaemolyticus* toward several antibiotics and heavy metals [69]. In this study, 2 GIs carrying heavy metal tolerance-related genes were identified, e.g., the GI 6 and GI 4 in *V. parahaemolyticus* L7-7 and N8-42 genomes, respectively. Additionally, several GIs carried

metabolism-related enzyme genes. For example, the GI 1 and GI 4 in *V. parahaemolyticus* N3-33 genome contained the *Vp N3-33_0631* and *Vp N3-33_2966* genes encoding a lipase, and a bifunctional GNAT family N-acetyltransferase/carbon-nitrogen hydrolase, respectively, which possibly benefited the bacterial survival in the environment.

There are approximately $10^{31}$ phage particles on Earth, considered the most abundant biological entity and an important driving force for bacterial evolution [70]. In this study, 5 prophage gene clusters (10,637 bp to 32,968 bp) were identified in *V. parahaemolyticus* N1-22, N4-46, N8-42, and Q8-15 genomes. They showed sequence similarity with the *Vibrio*_phage_K139, *Pseudomonas*_phage_D3, and *Vibrio*_phage_fs2 in *V. cholerae* K139, *P. aeruginosa* D3, and *V. cholerae* O139, respectively. The 5 prophage homologues contained a total of 156 genes, approximately 45.5% of which encoded unknown proteins. Notably, the *Vibrio*_phage_K139 homologue in *V. parahaemolyticus* N8-42 genome carried virulence determinants, such as the hemolysin (*Vp N8-42_2876*) and type II toxin-antitoxin system RelE/ParE family toxin (*Vp N8-42_2908*). Hemolysin is a well-studied virulence factor that can cause erythrocytolysis in the host [71]. Interestingly, the *Vibrio*_phage_K139 homologue (31,909 bp) was also identified in the *V. parahaemolyticus* N1-22 genome carrying a similar set of phage structure genes but different accessory genes, suggesting that *V. parahaemolyticus* N8-42 and N1-22 isolates may acquire the toxin genes from a common evolutionary ancestor by HGT with subsequent genome rearrangement into different *Vibrio* spp. and *V. parahaemolyticus* strains. Additionally, the *Pseudomonas*_phage_D3 homologue was identified in *V. parahaemolyticus* N4-46, and Q8-15 genomes, which provided the evidence at the genome level for the presence of extensive phage transmission between *V. parahaemolyticus* strains and even across species boundaries between *Vibrio* and *Pseudomonas* genera.

Integrons usually contain integrase (*intI*) genes, which can catalyze the merging or excision of gene cassettes through *attI* site-specific recombination [72]. They have been detected in clinically important pathogens and show a growing trend in environmental reservoirs [73]. The super integron was originally discovered in *V. cholerae* in 1999, which carried genes related to antibiotic resistance and pathogenicity [74]. In this study, 33 integrons (1082 bp to 39,523 bp) were identified in the six *V. parahaemolyticus* genomes, of which four super integrons were present in *V. parahaemolyticus* N4-46, N1-22, L7-7, and N3-33 genomes. Although approximately 23.4% of the integron-carrying genes (n = 435) encoded unknown proteins, the identified integrons might have a complex and diverse impact on the adaptability of the *V. parahaemolyticus* isolates to the environment and the host. For example, several integrons carrying virulence-related genes were found, such as the In 1, In 2, In 3, In 2, and In 1 in *V. parahaemolyticus* N1-22, L7-7, N3-33, N8-42, and Q8-15 genomes, respectively. Moreover, the super integron in *V. parahaemolyticus* N1-22 genome also carried toxic genes, e.g., type II toxin-antitoxin system RelE/StbE family mRNA interferase toxin (*Vp N1-22_3061*), and type II toxin-antitoxin system RelE/ParE family toxin (*Vp N1-22_3064*).

Previous research has indicated that the ISs belonging to IS3 and IS5 families played an important role in antibiotic resistance and virulence evolution in Gram-negative bacteria [75]. In this study, a few ISs were identified in *V. parahaemolyticus* L7-7, N3-33, N8-42, and Q8-15 genomes. However, no ISs carrying such genes were found, except that a *yoeB* gene (*Vp L7-7_4693*) encoding Txe/YoeB family addiction module toxin existed closely to the IS001 of the IS3 family in *V. parahaemolyticus* L7-7 genome.

CRISPR-Cas systems have been found to prevent the spread of plasmids and bacteriophages, and therefore limit the HGT of the MGEs [76]. They are a group of regularly spaced short palindromic repeats, which are composed of many short and conservative repeats and spacers. The CRISPRs and Cas proteins together form complete CRISPR-Cas systems, which are the most abundant defense component in archaea and bacteria [77]. In this study, six CRISPR-Cas gene clusters (97 bp to 825 bp) were identified in *V. parahaemolyticus* N1-22, N3-33, and Q8-15 genomes; however, all were incomplete ones lacking the Cas protein, which provided indirect evidence for possible active HGT mediated by the MGEs in the six *V. parahaemolyticus* isolates.

Previous studies have reported diarrhea cases caused by *V. parahaemolyticus* isolates lacking *tdh* and *trh* genes, suggesting that the other virulence-related factors exist and are attributable to the pathogenesis of *V. parahaemolyticus* [78]. In this study, a number of virulence-related genes (77 to 79) were identified in the six *V. parahaemolyticus* genomes originating in edible aquatic animals, e.g., *impI*, *vgrG*, *fliCDEFG*, *flrBC*, *yscOPQRST*, T3SS- and T6SS-related genes, which were involved in flagellar action, adhesion, colonization, biofilm formation, or epithelial cell invasion, suggesting possible health risks in consuming edible aquatic animals.

The global emergence and spread of MDR pathogenic bacteria poses an imminent threat to therapeutic options for human diseases [36]. AMP has been widely accepted as the first choice for the treatment of foodborne bacterial infections. However, it has been reported that the treatment efficiency for *Vibrio* spp. infection is low, perhaps because of the *crp* gene [79]. The inappropriate and continuous use of TET likely resulted in the increased resistance toward this drug [80]. In this study, approximately 17 to 20 antibiotic resistance-related genes were identified in the six *V. parahaemolyticus* genomes, including the *crp*, *tet (34)*, and *tet (35)* genes, consistent with their resistance phenotypes. Moreover, the other genes (e.g., *hns*, *acrB*, and *SoxR*) involved in resistance to fluoroquinolone, cephalosporin, and TET were also identified in the six *V. parahaemolyticus* isolates. Additionally, previous studies have linked heavy metal tolerance and antibiotic resistance in pathogenic bacteria [81]. In this study, some heavy metal tolerance-related genes (e.g., *cusASR*, *galT*, *zntA*) were also identified in the six *V. parahaemolyticus* isolates, which provided the evidence at the genome level for the co-resistance of potentially virulent *V. parahaemolyticus* isolates originating in edible aquatic animals.

## 5. Conclusions

This study was the first to specify genome features of six *V. parahaemolyticus* isolates recovered from three species of shellfish: *Paphia undulate*, *Perna viridis*, *Mactra veneriformis*; two species of fish: *Aristichthys nobilis*, *Carassius auratu*; and one species of crustacean: *Litopenaeus vannamei*, respectively. Most isolates with MDR phenotypes grew optimally at 3% NaCl and pH 8.5. Draft genome sequences of the *V. parahaemolyticus* isolates (4,937,042 bp to 5,067,778 bp) were determined using the Illumina Hiseq × 10 (Illumina, San Diego, CA, USA) sequencing technique. Comparative genome analyses revealed 4622 to 4791 predicted protein-encoding genes, of which 1064 to 1107 encoded unknown proteins. Various MGEs were identified, including GIs (n = 5 to 9), prophage gene clusters (n = 0 to 2), CRISPR-Cas systems (n = 0 to 6), integrons (n = 1 to 11), and ISs (n = 0 to 3). A number of antibiotic resistance-associated (n = 17 to 20) and virulence-associated (n = 77 to 79) genes, and strain-specific genes (n = 131 to 287) were also identified in the six *V. parahaemolyticus* genomes. The results demonstrated considerable genome variation mediated by the various MGEs in the six *V. parahaemolyticus* genomes. Overall, the results of this study fill gaps in our knowledge of genome evolution of *V. parahaemolyticus* isolates originating in aquatic animals. In future research, *V. parahaemolyticus* isolates of various aquatic animal origins should be investigated at the genome level for further understanding of the pathogenesis mechanisms and epidemiological features of the leading foodborne sea pathogens worldwide.

**Supplementary Materials:** The following supporting information can be downloaded at: https://www.mdpi.com/article/10.3390/d14050350/s1, Table S1: Oligonucleotide primers used in this study; Table S2: The identified GIs in the six V. parahaemolyticus genomes; Table S3: The identified prophages in the V. parahaemolyticus genomes; Table S4: The identified ISs in the V. parahaemolyticus genomes; Table S5: The identified repeats at the end of scaffolds of the V. parahaemolyticus genomes; Figure S1: The k-mer analysis for V. parahaemolyticus subread data based on the number of unique 17-mers. A to F: V. parahaemolyticus L7-7, N1-22, N3-33, N4-46, N8-42, and Q8-15 genomes, respectively.

**Author Contributions:** D.X.: investigation, data curation, and writing—original draft preparation; X.P. and L.X.: discussion and supervision; L.C.: funding acquisition, conceptualization, and writing—review and editing. All authors have read and agreed to the published version of the manuscript.

**Funding:** This study was supported by Shanghai Municipal Science and Technology Commission, grant number 17050502200, and National Natural Science Foundation of China, grant number 31671946.

**Institutional Review Board Statement:** Not applicable.

**Informed Consent Statement:** Not applicable.

**Data Availability Statement:** Draft genome sequences of the six *V. parahaemolyticus* isolates have been deposited in GenBank database under the accession numbers: JAHRCZ000000000, JAHRDA000000000, JAHRDB000000000, JAHRDC000000000, JAHRDD000000000, and JAHQCP000000000.

**Acknowledgments:** The authors are grateful to Meng Sun and Kaiyue Zhang for their help in the manuscript preparation.

**Conflicts of Interest:** The authors declare no conflict of interest.

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
