# Peer review of "Survival and Genome Diversity of Vibrio parahaemolyticus Isolated from Edible Aquatic Animals"

_diversity, doi:10.3390/d14050350_

Round 1

Reviewer 1 Report

Submitted manuscript Diversity-1665625 entitled as “Survival and genome diversity of Vibrio parahaemolyticus isolates originating in edible aquatic animals” is well analyzed piece of work and written properly. Here author has done the comparative analysis of six Vibrio species isolated from the edible aquatic animals. Here are some suggestions which need to be either answered or done the amendments.

  1. In title word “originating” make the confusion that mentioned Vibrio isolates are peculiar host of these species Paphia undulate, Perna viridis, Mactra veneriformis; Aristichthys nobilis, Carassius auratu; Litopenaeus vannamei. Hence isolated would be more logical.
  2. What is the pathogenicity information of these Vibrio strain, any supportive genomic data will definitely add on the value of the paper?
  3. I would recommend that method of the paper for in silico work should be in detail.
  4. As far as ABC transporters superfamily is concerned, it is having some potent family members which need to be discussed here in details. You can take the help of Kumar et al Pathogens 2021, 10(9), 1127; https://doi.org/10.3390/pathogens10091127 and cite.
  5. Figure 4, For phylogenetic tree, used algorithms and bootstrap value should be in legends.
  6. Figure 3 and 7 - color coding of genome map should be more descriptive for better understanding of readers.
  7. Take care of the English and grammar more carefully.

All the best.

Author Response

Thank you!

Reviewer 2 Report

Review of manuscript entitled Survival and genome diversity of Vibrio parahaemolyticus isolates originating in edible aquatic animals by Xu et al. for Diversity

[I've also included these comments in pdf form in case this renders weird]

I am recommending that this paper be rejected unless the major issues regarding the genome assembly analyses and phylogenetic analyses are addressed. This manuscript has fundamental problems—it is missing many pieces that should be expected from papers reporting genomes and utilizing phylogenies.

Below are the issues—all of these need to be addressed in the text of the paper.

Phylogenies

How were supports calculated on trees? Were they calculated via bootstrap or jackknife or something else? How many replicates were used if they were based on resampling?

There may be an issue of long branch attraction—similar length branches and short internodes near the root of the tree may be indicative of this. I suggest inferring a tree with the maximum likelihood optimality criterion (in a program like PhyML or RAxML with appropriate supports calculated) to corroborate the existence of these clades and help to rule of long branch attraction (maximum likelihood is more robust to long branch attraction).

Genome assembly analysis

Assemblies are incomplete – could genetic elements have been missed? If so, how many do expect could have been missed?

Are any genetic elements located at the end of scaffolds or contigs? This may shed some light on the possibility of the assembly missing some genetic elements.

What is the coverage depth of the assembly (on average)? How much does it vary? To what degree does it taper at the end of scaffolds or contigs (if it does taper)?

Paired-end sequencing was used—what was the insert size? What proportion of read pairs map concordantly? Do you speculate that the unassembled gaps between scaffolds were comprised of repetitive DNA?

All of these questions need to be addressed or answered in some way in the text of the paper for it to be accepted for publication. These methodological issues and omissions are fundamental.

Author Response

Thank you!

Round 2

Reviewer 2 Report

The authors have done a great job addressing my reservations. I recommend this manuscript for publication. I look forward to seeing this paper published in Diversity.

I somehow missed the mention of the read depth on my earlier read and review of the manuscript (it was in an obvious place). This was my mistake and I apologize.

Author Response

Dear Reviewer 2,

Thank you very much for your encouragement to our revision work!

No mention of the sequencing read depth at all. We apologize for the very slow speed of uploading the revised files online last time. The Supplementary Material: Figure S1 has been uploaded, together with the revised manuscript Diversity-1665625R2. Your comments and suggestions are very valuable and helpful for improving our manuscript (Diversity-1665625), and also have important guiding significance to our researches. We look forward to next collaboration with you in the near future.

Once again, many sincere thanks for your great help to our manuscript (Diversity-1665625) for the publication in Diversity

Yours sincerely,

The authors:

Dingxiang Xu, Lanming Chen*

College of Food Science and Technology, Shanghai Ocean University, Shanghai 201306, China.

Xu Peng

Department of Biology, University of Copenhagen, Copenhagen N, Denmark.

Lu Xie

Institute for Genome and Bioinformatics, Shanghai Institute for Biomedical and Pharmaceutical Technologies, Shanghai 201203, China